# Development of Combined Methods Using Non-Destructive Test Methods to Determine the In-Place Strength of High-Strength Concretes

**Tuba Demir** [1],*, **Muhammed Ulucan** [2] and **Kürşat Esat Alyamaç** [2]

1   Civil Engineering Department, Technology Faculty, Firat University, 23119 Elazig, Turkey
2   Civil Engineering Department, Engineering Faculty, Firat University, 23119 Elazig, Turkey
*   Correspondence: t.demir@firat.edu.tr

**Abstract:** This study aims to develop combined methods with dual and triple use of different non-destructive test (NDT) methods and to examine the effectiveness of these methods. For this purpose, nine different high-strength concrete series were produced, and destructive and NDT methods were applied to these samples on the 3rd and 90th days. Surface hardness, ultrasonic pulse velocity, and penetration resistance were considered from NDT methods. Analyses were made on the response surface method using the NDT measurements and compressive strength values obtained, and four different mathematical models were developed. Models 1, 2, and 3 were designed with dual use of NDT methods, and model 4 was designed with triple use. The absolute relative deviation values for all the developed models' early and final-age strengths were below 10%. It is of great importance to determine concrete quality with high accuracy and practicality, especially in places like Elazig, where there are thousands of newly constructed buildings since the Sivrice-Elazig earthquake, and rapid production is required. Therefore, it is thought that determining the strength values with high accuracy using the developed combined methods without damaging the building element will provide benefits in terms of time, labor, and cost.

**Keywords:** highstrength concrete; compressive strength; non-destructive test methods; combined methods; response surface method





## 1. Introduction

### 1.1. Literature Review

High-strength concrete (HSC) is a special variety of concrete that offers innovative applications to the mechanical properties of reinforced concrete structures [1]. In order to produce these concretes and obtain better mechanical properties, selecting the components appropriately and using a low water-to-cement (W/C) ratio is necessary. Aitcin et al., in their study, emphasized that the quality of concrete components is of great importance to obtaining HSC [2]. Portland cement is widely used in HSC production, as aggregate, basalt, quartz, granodiorite, andesite, etc. are used [3].

HSCs are special concretes that are extremely durable and have advanced mechanical properties. Many HSC applications in the United States, Germany, Canada, France, and Turkey have demonstrated the advantages of this concrete type [4]. However, the project's duration is one of the most important limiting conditions when constructing these structures. The project must be completed in the targeted time, before the start of production. When the project period is exceeded, significant additional costs and major financial problems arise [5]. One of the most important issues affecting this time is the time required by the concrete to gain strength. The speed of the project is significantly dependent on the formwork stripping times. The parameter that directly affects this speed is the concrete compressive strength. In particular, the early age of concrete compressive strength plays a major role in determining the formwork stripping times. For this reason, it

is determined by various tests whether the concrete reaches the targeted strength or not. These determination studies are carried out by two methods [6,7]. In the first method, a sample is taken from the concrete during casting, and the concrete compressive strength is determined by following the relevant procedures. In the second method, after the concrete is placed and set, the concrete compressive strength is predicted with the help of non-destructive test (NDT) methods [8,9]. Specifically, it is very important to determine the in-place strength of concrete, to ensure the safety of reinforced concrete structures [10,11]. Both destructive test (DT) and NDT methods are used to determine the concrete strength of existing buildings [12–14]. Although there are many studies on DT methods, few studies are conducted on NDT methods in HSCs.

One of the most important challenges related to the evaluation of structures in the construction industry is the determination of concrete quality and performance. Concrete evaluation methods are explained in the standards of many countries, such as the European Standard, ACI, and TSE [15,16]. These evaluation methods are DT and NDT test methods. In the DT method, core samples are taken from the carrier elements, and strength values are determined. However, this method is expensive and time-consuming, and damages the carrier element. NDTs are a great alternative to eliminate these disadvantages and to determine the strength values in an economical, practical, and easier way, without damaging the carrier elements [17,18]. The NDT method provides many conveniences regarding time, cost, and labor. The NDT method provides the opportunity to evaluate the durability, homogeneity, internal structure, and strength properties of the concrete [19]. In addition, NDTs are applied without damaging old or new structures [20].

In construction, concrete samples are usually taken to determine the early- and final-age strengths. It is not technically logical nor effective to take concrete samples for all ages and for many time periods and to determine the strength with the help of these samples, because taking a large number of concrete samples means loss of time, increase in cost, and additional labor. NDT methods are used to avoid these problems, and to determine and control the compressive strength of concrete whenever desired [21]. In the evaluation of existing structures, NDT techniques are applied in a combined manner. In this method, tests are carried out without disturbing the functionality of the part or system. The surface hardness and ultrasonic pulse velocity method are the most widely used test methods in the literature and in practice [22]. Using a single test method in NDT methods gives approximate results and is insufficient [23]. Therefore, it should be evaluated by making measurements with different NDT methods. Evaluating the advantages and disadvantages of each test method gives more acceptable results, as each has its advantages and disadvantages. Being practical, economical, and predictive of the compressive strength of concrete in the desired age range are the common advantages of all of them [23]. However, the common disadvantage of all of them is that the amount of deviation of the results obtained after the tests can be higher than expected when they are used alone [8]. In order to eliminate this disadvantage, there are combined methods developed using surface hardness and ultrasonic pulse velocity methods in the literature [24,25]. Studies have shown that the results of these combined methods give higher accuracy than the results of individual test methods [26].

In previous studies, there were limited studies using combined NDT methods to determine the strength of HSCs [27]. Among these studies, A. Lamproulos et al., used the ultrasonic pulse velocity and surface hardness NDT methods for ultra-high-strength concretes. The main purpose of his studies was to determine the strength properties of concrete by testing the dual combination of NDT methods and to evaluate the reliability of these methods. To this end, they prepared different mixtures and performed the tests they determined. They made comparisons with the data they obtained and developed a mathematical model. As a result, the researchers stated that the compressive strength of concrete could be determined with a lower error rate by using multiple NDT methods [8]. However, the model obtained by the researchers was limited only to steel fiber ultra-high-strength concretes prepared with the mixing ratios they determined. Another

study in the literature estimated concrete strength using NDT methods such as surface hardness and penetration resistance methods. In this study, unlike other applications, the penetration resistance method was used. As a result of the analysis, a combined method was developed to estimate the compressive strength of concrete. They stated that the model obtained as a result of the study provides a prediction in the determination of compressive strengths in a limited time in urban transformation projects. They emphasized that this method can also be developed for determining early- and final-age strengths by conducting experimental studies [28].

### 1.2. Main Objective and Scope of the Study

NDT methods are very useful in the rapid evaluation of structures. These test methods allow measurements to be made in shorter time and more easily, without damaging the structure. They can be used to locate and estimate the size of cracks and voids in concrete [18]. In addition, long-term changes in concrete can be monitored. This study aimed to develop combined NDT methods that predict early- and final-age strength with high accuracy. There are few studies in the literature on the determination of in-place compressive strength of HSCs. In existing studies, ultrasonic pulse velocity and concrete surface hardness tests are commonly used NDT tests. However, a specific mixture design has not been presented with the data obtained from these tests. This study aims to develop double and triple combined methods by considering surface hardness, ultrasonic pulse velocity, and penetration resistance methods. For this purpose, statistical analyses on the response surface method (RSM) were made using the obtained measurement results and strength values, and combined methods were developed. The developed combined methods were compared among themselves and with the experimental results, and their effectiveness was evaluated in detail. Then, the advantages and disadvantages of each of these combined methods were examined, and solution suggestions were presented. It was observed that the developed combined methods predict both early- and final-age strength values with high accuracy. Using these combined methods will likely provide significant benefits in terms of time, cost, and labor. In addition, this study aims to lead other studies on this subject by making a new contribution to the literature with the mathematical model to be developed. The flow chart of the study is given in Figure 1.

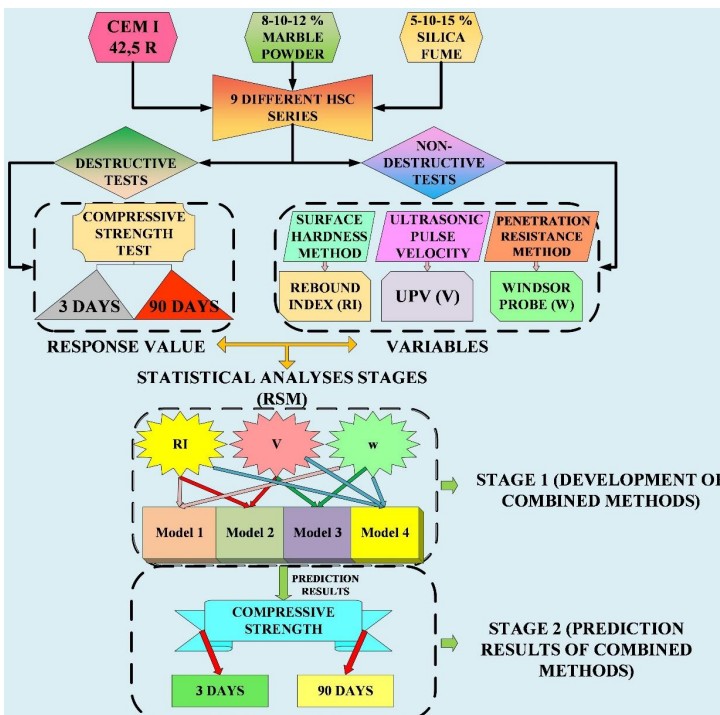

**Figure 1.** Flow chart of the presented study.

## 2. Experimental Investigation and Test Methods

### 2.1. Materials

In this study, CEM I 42.5 R Portland cement, produced by TS EN 197-1 standard and supplied from the Elazig Seza cement factory, was used as the binder [29]. Locally, silica fume was obtained from Antalya Eti metallurgy, and marble powder was obtained from the Elazig organized industrial zone. A pozzolanic activity test was carried out for silica fume. As a result of the experiment, the pozzolanic activity index was determined [30]. The marble powder used in the concrete mixture was used to create a filler effect. In order to determine the usage ratio of marble powder in concrete, many trial mixtures were cast, and the optimum usage ratio was determined [31]. The physical and chemical properties of HSC components are presented in Table 1.

**Table 1.** Chemical and physical properties of cement, silica fume, and marble powder (%).

| Chemical Properties | Cement (C) | Silica Fume (SF) | Marble Powder (MP) |
|---|---|---|---|
| CaO | 63.19 | 0.40 | 40.45 |
| $SiO_2$ | 19.07 | 94.10 | 28.35 |
| $Fe_2O_3$ | 3.72 | 1.50 | 9.70 |
| $Al_2O_3$ | 4.82 | 0.90 | 0.17 |
| $SiO_3$ | 2.94 | 94.10 | 0.02 |
| $Na_2O$ | 0.39 | 0.40 | 0.05 |
| $K_2O$ | 0.62 | 0.90 | 0.01 |
| MgO | 1.83 | 0.10 | 16.25 |
| Cl | 0.01 | - | - |
| Insoluble residue | 0.56 | - | - |
| Loss of ignition | 3.43 | - | 4.84 |
| **Physical Properties** | | | |
| Specific surface | 3838 | | 3920 |
| Specific gravity (g/cm$^3$) | 3.13 | 2.20 | 2.71 |
| Initial setting time (min) | 135 | - | - |
| Final setting time(min) | 215 | - | - |
| Total volume exp. (mm) | 1 | - | - |

Aggregates with high strength and compactness are used in HSCs. Necessary aggregate tests were carried out for the aggregates used in this study, and their suitability for use in HSCs was determined. As a result of the experiments, andesite aggregate with high density and strength was selected [32]. This aggregate contains 52–63% quartz. It is suitable for HSCs due to its dark color, and its non-absorbent, non-dispersive, and dense texture [33]. The maximum aggregate particle size used in experimental studies is 16 mm. The aggregates were divided into 4 different groups: fine-1, (0–2) mm; fine-2, (2–4) mm; coarse-1, (4–8) mm, and coarse-2, (8–16) mm. The aggregates used in the experimental study are shown in Figure 2.

### 2.2. Mix Design and Tests on HSC

In the first stage of the study, many trial mix castings were made. Then, a large number of reference concrete samples with different mixing ratios were prepared [34]. Firstly, the compressive strength test, which is one of the destructive tests, was applied to these samples. According to the data obtained from these tests, the concrete series to which the non-destructive tests would be applied were determined. NDT tests were applied to these series, and the accuracy of the results was compared. In this way, studies with many stages were carried out.

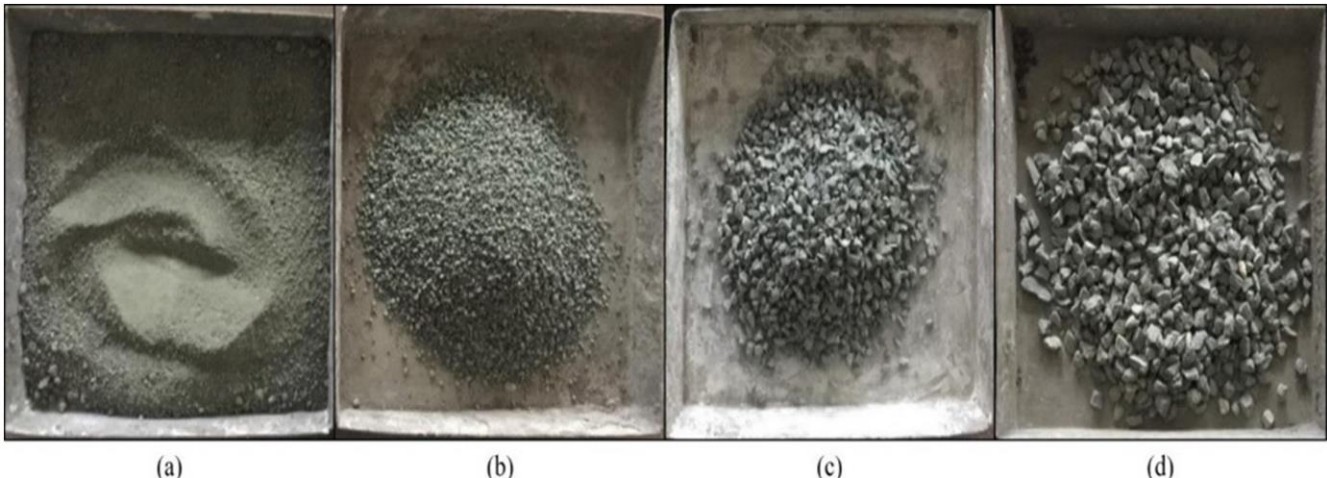

**Figure 2.** Aggregate grain diameters (**a**) (0–2) mm, (**b**) (2–4) mm, (**c**) (4–8) mm, (**d**) (8–16) mm.

In the second stage of the study, mixture designs were determined to analyze and interpret the results of both DT and NDT methods of concrete mixtures containing different amounts of cement dosage, silica fume, and marble powder. For this purpose, 9 different concrete series were produced. Mixture components and amounts of the prepared concrete mixtures are given in Table 2. In the series' naming, the first letters of 'HSC' indicate high-strength concrete, the next number is the cement dosage, and the serial number is the last digit. Both DT and NDT methods were applied to the produced series on the 3rd and 90th days. Among the NDT methods, the surface hardness, ultrasonic pulse velocity, and penetration resistance were considered. Analyses were made using the obtained compressive strength and NDT results.

**Table 2.** Mixture amounts of high-strength concrete.

| Mix ID | Cement | Water | Silica Fume | Marble Powder | Fine-1 | Fine-2 | Coarse-1 | Coarse-2 | Chemical Additive |
|--------|--------|-------|-------------|---------------|--------|--------|----------|----------|-------------------|
| HSC400-1 | 360 | 80 | 28.1 | 103 | 559 | 368 | 406 | 609 | 11.2 |
| HSC400-2 | 380 | 100 | 14.1 | 101 | 550 | 362 | 400 | 599 | 11.8 |
| HSC400-3 | 380 | 120 | 14.1 | 79 | 547 | 360 | 389 | 583 | 11.8 |
| HSC450-1 | 383 | 90 | 47.4 | 78 | 544 | 358 | 387 | 580 | 11.9 |
| HSC450-2 | 383 | 113 | 47.4 | 114 | 504 | 332 | 374 | 562 | 11.9 |
| HSC450-3 | 405 | 135 | 31.6 | 74 | 516 | 340 | 367 | 550 | 12.6 |
| HSC500-1 | 475 | 100 | 17.6 | 116 | 513 | 338 | 381 | 572 | 14.7 |
| HSC500-2 | 450 | 100 | 35.1 | 95 | 517 | 341 | 376 | 564 | 14.0 |
| HSC500-3 | 425 | 150 | 52.7 | 87 | 473 | 312 | 344 | 516 | 13.2 |

*2.3. Response Surface Method*

The RSM is a widely used method in a wide range of fields, such as civil engineering, chemical engineering, and materials engineering, which determines the relationship between parameters and responses [35]. It is mostly used in concrete mix ratio designs in civil engineering [35,36]. The main purpose of this method is to determine the relationship between parameters and responses with fewer tests, to determine the best response value [37]. RSM-based design consists of Central Composite Design (CCD) and Box-Behnken Design (BBD) types [38]. In studies using CCD, there are generally 5 different levels of independent variables. The BBD method, on the other hand, is widely used when the independent variables have 3 different levels. BBD is a more economical design, as it requires less trials than CCD. However, performing fewer experiments reduces the accuracy of the predictions of these regions in optimization calculations [39–41].

### 2.4. Non-Destructive Test Methods

NDT is widely used for the determination of different properties of many building materials. This method allows a detailed evaluation of the strength characteristics of existing reinforced concrete structures [42]. In addition, using this method, many properties of concrete such as void structure and durability can be determined. Surface hardness and ultrasonic pulse velocity are among the commonly used NDT methods [43]. In this study, surface hardness, ultrasonic pulse velocity and penetration resistance methods were considered.

#### 2.4.1. Surface Hardness Method

One of the most widely used NDT methods is the surface hardness method. This method gives an approximate idea of concrete strength by making use of the surface hardness of concrete [25]. While applying this method, an instrument known as a Schmidt test hammer is used. This test hammer was developed in 1948 by the Swiss engineer Ernst Schmidt to measure the hardness of concrete [44]. This test method is widely used around the world, but it cannot give high-accuracy results on concrete strength alone. In this method, 11 different measurement values are taken on the surface of the produced concrete samples. The largest and smallest rebound values obtained as a result of the measurements are neglected. The test result is determined by taking the arithmetic average of the remaining 9 measurements [45]. Test standards are carried out as specified in TS EN 12504-2 or DIN 1048-1 [16,45]. The surface hardness method applied on the concrete sample is shown in Figure 3a.

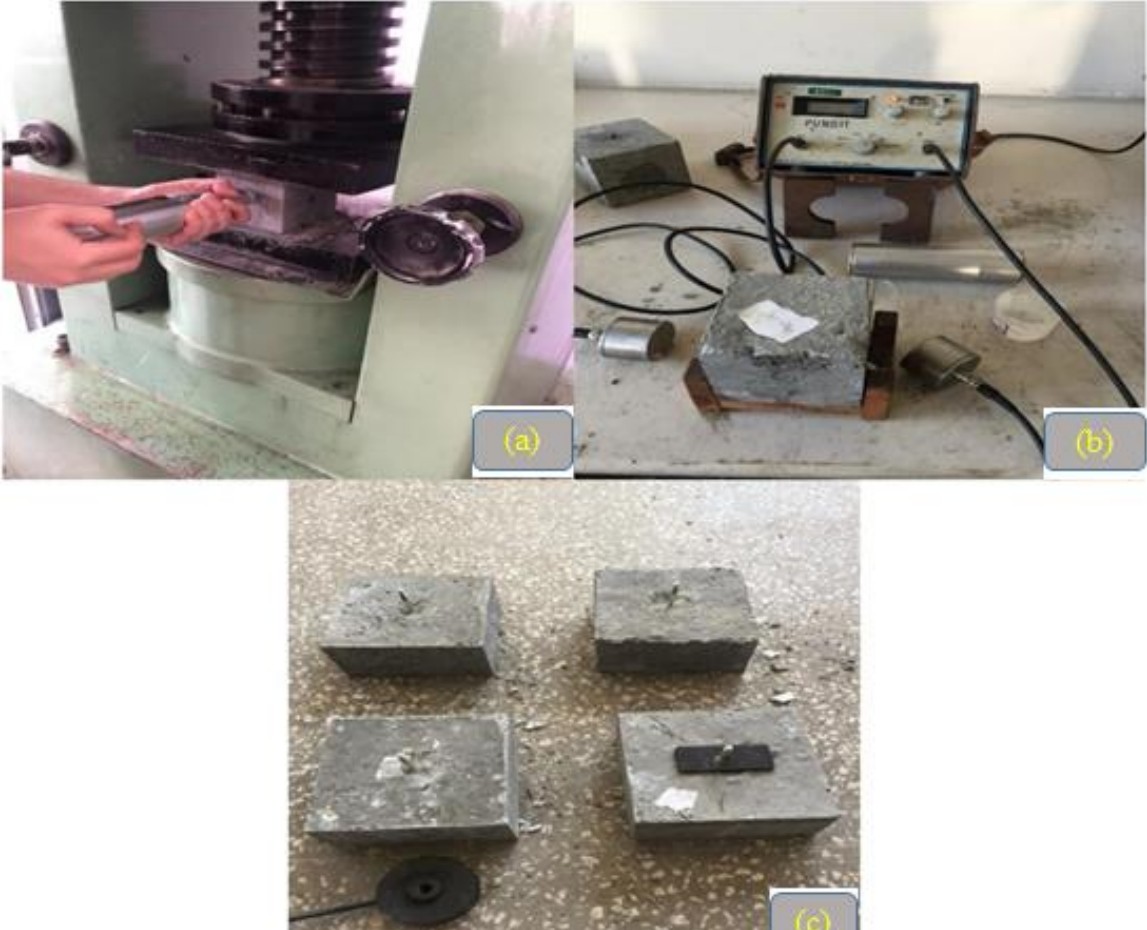

**Figure 3.** Applied non-destructive tests on concrete samples: (**a**) Surface hardness (**b**) Ultrasonic pulse velocity. (**c**) Penetration resistance.

### 2.4.2. Ultrasonic Pulse Velocity

The ultrasonic pulse velocity test is used to measure the propagation speed of ultrasonic waves in concrete. The wave of axial pulses is generated by an electro-acoustic transducer (transmitter and receiver) in contact with the concrete surface under test. The vibration wave traveling a known distance in concrete is converted into electrical signals by a second transducer, and the speed of the wave is measured by the electronic timing circuit. This pulse velocity depends on the type of aggregate, the number of voids in the concrete, the crack state, the age of the concrete, and the water content [24]. Test standards are detailed in TS EN 12504-4 and ASTM C 597 [46–48]. Ultrasonic pulse velocity test equipment is shown in Figure 3b.

### 2.4.3. Penetration Resistance Method

The penetration resistance method was performed with Windsor probe testing equipment by ASTM C803 [49]. Particular samples of 200 × 200 × 100 mm were produced for this test. A penetration test was applied to each sample. This test is applied in two ways, at low (d) and standard (s) energy. In the low-energy penetration experiment, the metal probe placed inside the Windsor probe is pushed 5 cm inside the barrel. In the case of testing with standard energy, the metal probe is not pushed into the barrel. The applied Windsor probe tests were made at the standard energy level. The penetration test result is the length of the part of the metal probe outside of the concrete [50]. As the concrete strength increases, the length of the outer part of the probe also increases. The test method applied is shown in Figure 3c.

## 3. Results and Discussions

### 3.1. NDT Results and Hardened Concrete Properties

The 9 different HSC series were removed from the curing pool on the 3rd and 90th days, and NDT measurements were applied. First, surface hardness readings were performed on the samples, followed by the ultrasonic pulse velocity and penetration resistance methods. After NDT measurements, compressive strength tests were applied, and strength values were determined. The produced concrete series, strength values, and NDT measurements are presented in Table 3: Surface hardness readings are given as rebound index (RI), ultrasonic pulse velocity values as (V), and penetration resistance test results were named Windsor probe (w).

**Table 3.** Values of destructive and non-destructive test results.

| Mix ID | $f_c$,3 Days (MPa) | $f_c$,90 Days (MPa) | RI | | V (km/s) | | w (mm) | |
|---|---|---|---|---|---|---|---|---|
| | | | 3-Days | 90-Days | 3-Days | 90-Days | 3-Days | 90-Days |
| HSC400-1 | 53.6 | 70.9 | 23 | 30 | 5.06 | 5.36 | 52.3 | 55.0 |
| HSC400-2 | 56.5 | 78.7 | 25 | 33 | 5.31 | 5.43 | 55.3 | 58.8 |
| HSC400-3 | 54.2 | 75.8 | 24 | 32 | 5.13 | 5.37 | 53.0 | 57.7 |
| HSC450-1 | 56.3 | 79.1 | 26 | 33 | 5.29 | 5.46 | 55.1 | 60.2 |
| HSC450-2 | 58.6 | 83.3 | 28 | 35 | 5.38 | 5.59 | 57.2 | 62.4 |
| HSC450-3 | 55.8 | 77.1 | 25 | 32 | 5.25 | 5.41 | 54.5 | 58.5 |
| HSC500-1 | 56.7 | 81.4 | 26 | 34 | 5.32 | 5.52 | 55.5 | 61.3 |
| HSC500-2 | 61.7 | 88.2 | 30 | 37 | 5.42 | 5.66 | 58.7 | 64.3 |
| HSC500-3 | 55.9 | 81.8 | 25 | 32 | 5.26 | 5.54 | 54.4 | 61.4 |

Figure 4 shows the effect of the results of the applied NDT methods on the strength. Figure 4a,d show the effect of the *RI* values obtained after the surface hardness tests on $f_c$,3 and $f_c$,90 days. As the strength value of the produced concrete samples increased, an increase was observed in the *RI* values of the test hammer. Similar results were obtained in other studies [51]. A detailed examination of these graphics clearly shows that the estimated values for $f_c$,3 days were better than those for $f_c$,90 days. However, this difference indicated

that the surface hardness test alone gave estimated results, and its use with another NDT method will increase the accuracy.

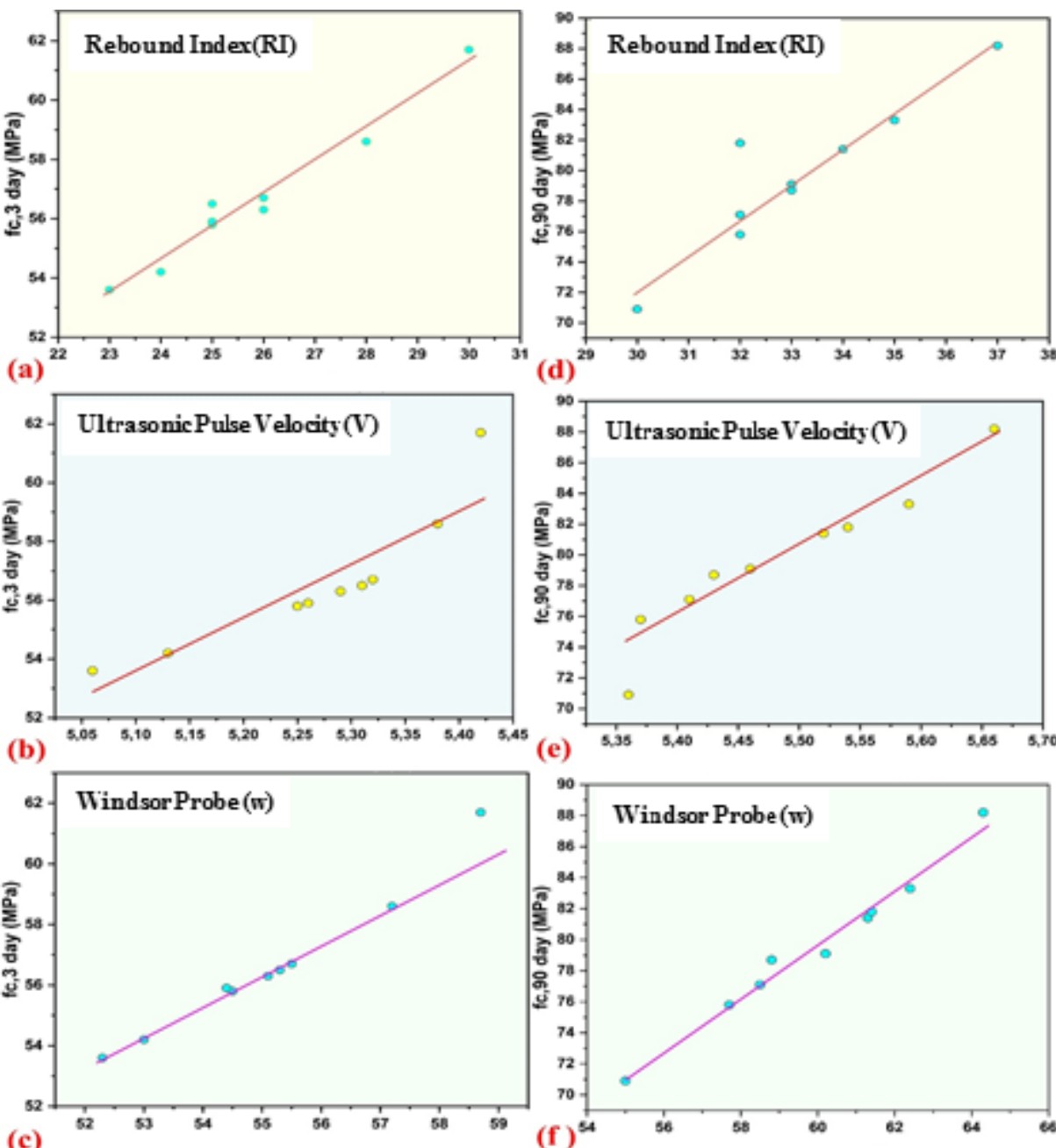

**Figure 4.** Relationship between non-destructive test method results and compressive strength.

Figure 4b,e show the relationship between ultrasonic pulse velocity (*V*) test results on strength. In this test method, the transition rate was obtained by dividing the length of the sample by the transit time. Therefore, as the strength of the concrete increased, the transition time shortened, and *V* increased. The test result measurements obtained also confirm this. Similar results were obtained in other studies [52]. A detailed examination of the Figures shows that there were approximate results between the test results and the strength values. One of the main reasons for this was the sensitivity of this test method to moisture content. Therefore, using this test method alone gave approximate values, and increasing the degree of accuracy with its combined use will provide significant benefits.

Figure 4c,f show the effects of Windsor probe test results on strength. A detailed examination of the results of this test method showed that, as the Windsor probe values increaseD, the strength values increased. In samples with high compressive strength, since the probe sAnk less into the concrete, the length of the outer part of the probe also increased. In addition, it was seen that the relationship between the 3- and 90-day test results and *w* gave higher-accuracy results than other test methods. The main reason for this situation was that the Windsor probe test gave an idea about the internal structure of the concrete, so more positive results were obtained.

### 3.2. Designing RSM-Based Combined Methods

The main purpose of this section is to develop combined methods that estimate the approximate strength values with higher accuracy due to the single use of NDT methods. To this end, mathematical models were developed using more than one NDT method, with the aim of predicting the strength values with high accuracy.

### 3.2.1. Early-Age Concrete Strength

Early-age strength significantly affects the formwork stripping times and, indirectly, the project duration in a building. For this reason, it is of great importance to measure the early-age concrete strength with high accuracy and without damaging the building. This study aimed to develop mathematical models that predict the early-age strength with high accuracy without damaging the building, using NDT methods. While designing the models, double and triple combined methods were developed using different NDT methods. Developed models and considered NDTs are shown in Table 4.

**Table 4.** Developed models and considered non-destructive test methods.

| Models | Surface Hardness (R) | Ultrasonic Pulse Velocity (V) | Penetration Resistance (w) |
|---|---|---|---|
| Model 1 | ✓ | | ✓ |
| Model 2 | ✓ | ✓ | |
| Model 3 | | ✓ | ✓ |
| Model 4 | ✓ | ✓ | ✓ |

Models 1, 2, and 3 were developed with dual uses of NDT methods, and model 4 was developed with triple uses. Two- and three-dimensional graphics showing the developed binary models and their effects on early-age strength are shown in Figure 5. Figure 5a shows the effect of the test methods considered on the strength of model 1. Figure 5a reveals that the strength values increased as the *R* and *w* values increased. The main reason for this situation was that, as the strength increased, the rebound in the concrete test hammer increased, and since the probe sank into the concrete less, the length of the outer part of the probe increased. Figure 5b shows the effect of the test methods considered on the early-age strength of model 2. This graphic clearly reveals that the strength values increased as the *R* and *V* values increased. As the strength values increased, the void ratio of the concrete decreased, and the transition time became shorter. Figure 5c shows the effect of *V* and *w* values on the early-age strength of model 3. Similar results were obtained here, and it was clearly demonstrated that as *V* and *w* values increased, the early-age strength values increased.

The two-and-three-dimensional effect graphics of model 4, which was developed by using three NDT methods together, are shown in Figure 6. A detailed examination of the Figures reveals that the early-age strength values increased as the *R*, *w*, and *V* values increased. It was seen that the early-age strength values of the concrete series produced all vary between 50–65 MPa. The values given for the variables on the side of the Figures represent the arithmetic mean of these test results' lowest and highest values.

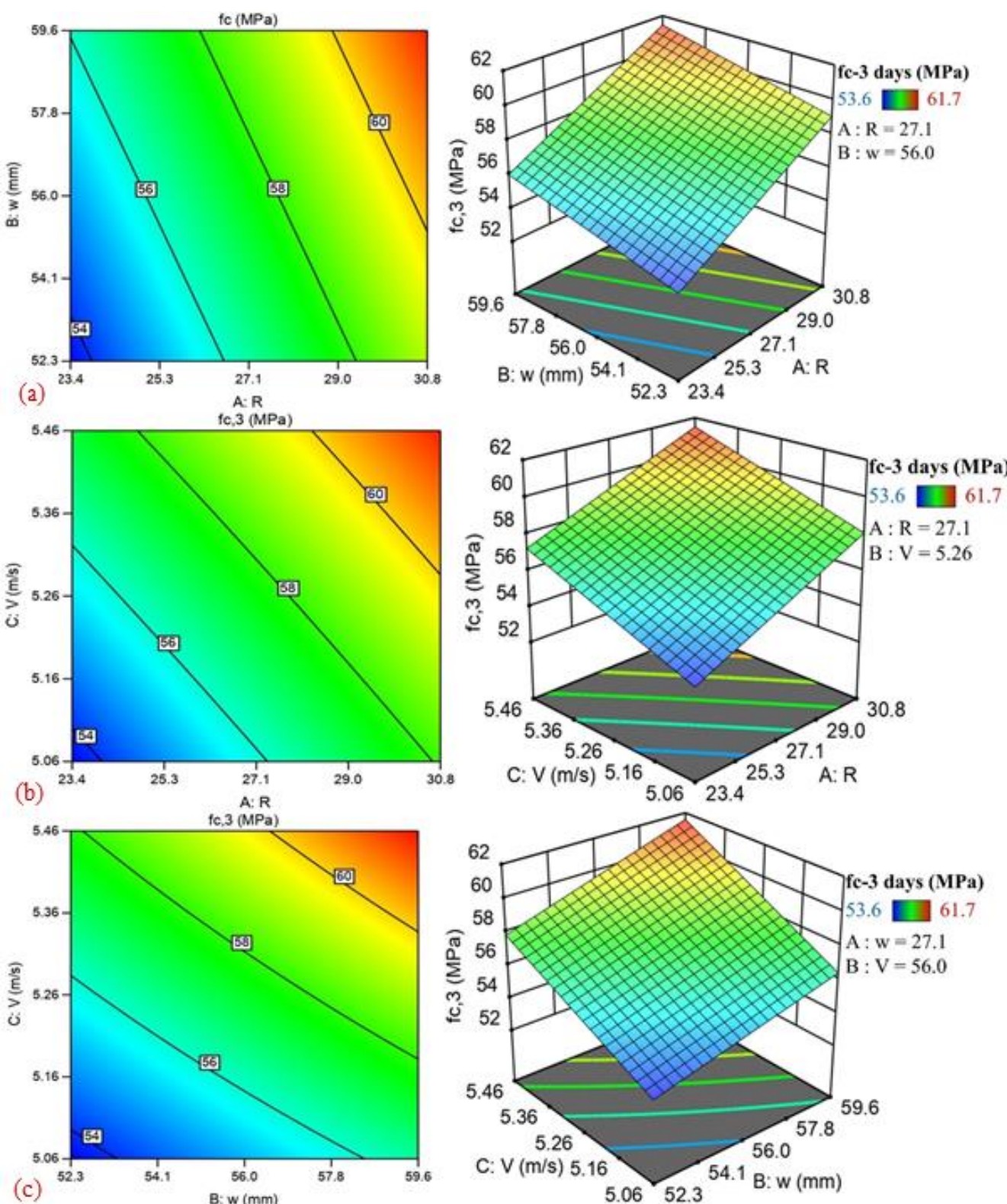

**Figure 5.** The effect of developed models on early-age strength: (**a**) Model 1, (**b**) Model 2, and (**c**) Model 3.

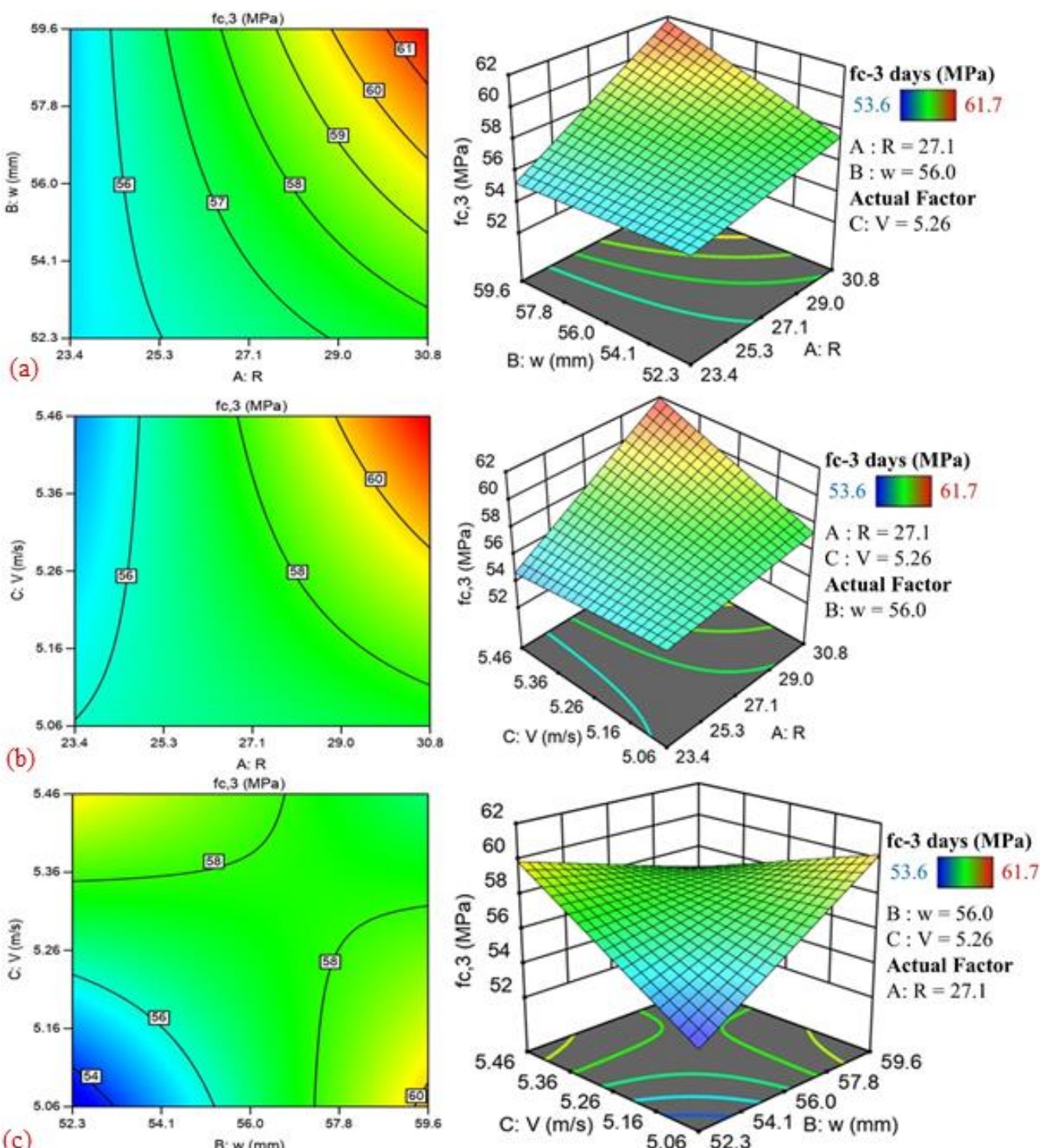

**Figure 6.** The effect of model 4 on early-age strength: (**a**) Model 1, (**b**) Model 2, and (**c**) Model 3.

### 3.2.2. Final-Age Concrete Strength

Final-age strength is of great importance in evaluating the quality and performance of concrete. Four mathematical models were developed to predict the final-age strength with high accuracy.

Figure 7 shows the effect on the final-age strength of the developed models by the dual use of NDT methods. Figure 7a reveals the outputs of the parameters considered in model 1 developed to estimate the final-age strength. As the strength values increased, the rebound value of the concrete test hammer and the length of the outer part of the Windsor

probe increased. Figure 7b illustrates the increase in *R* and *V* values as the final-age strength values increased. This situation resulted in an increase in the *V* values due to the decrease in the void ratio and the shortening of the transition time. Figure 7c, on the other hand, shows that *V* and *w* values increased with the increase in final-age strength.

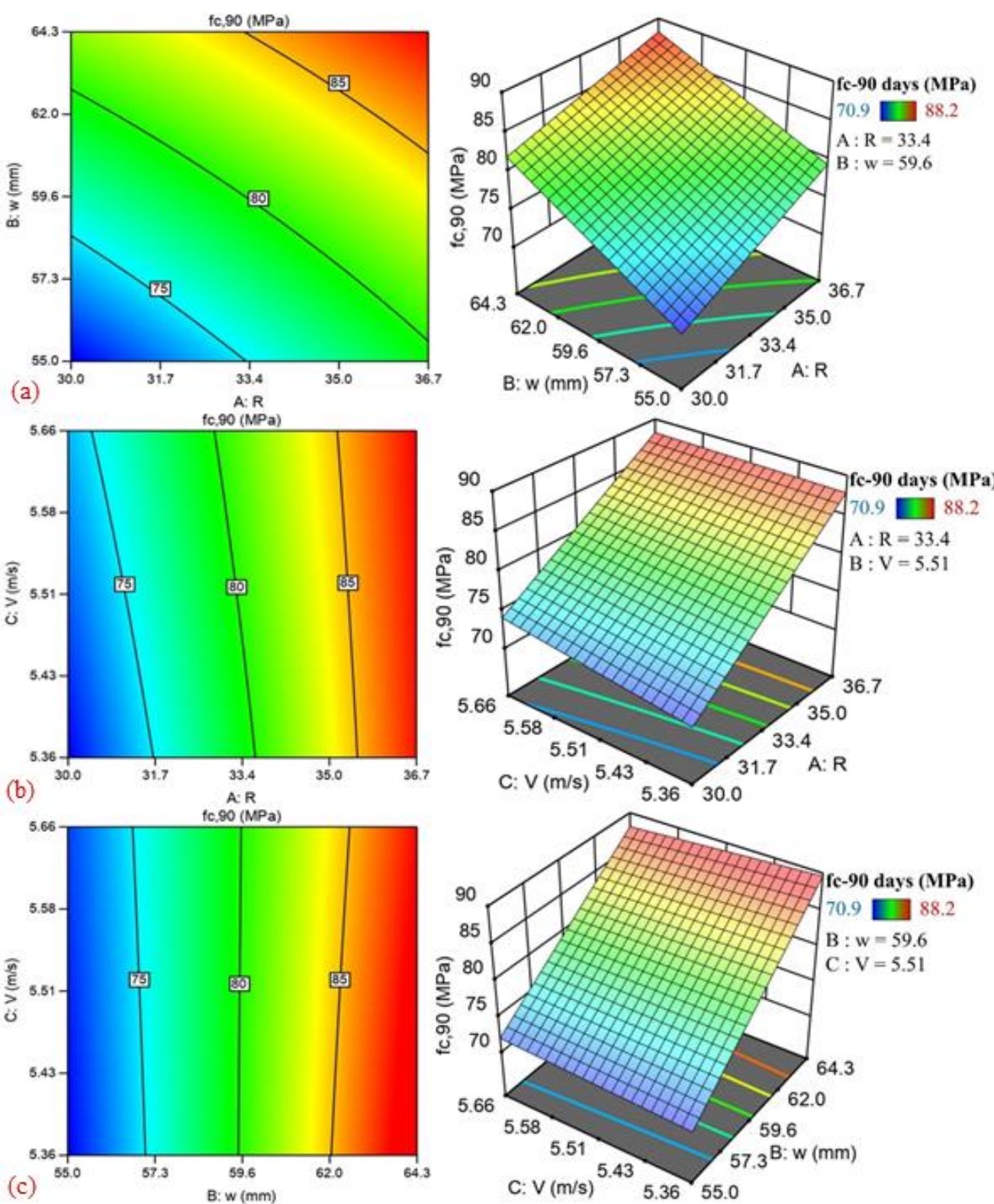

**Figure 7.** The effect of developed models on final-age strength: (**a**) Model 1, (**b**) Model 2, and (**c**) Model 3.

The graphics for model 4, which was developed by using three NDT methods together to predict the final-age strengths with high accuracy, are shown in Figure 8. A detailed evaluation of Figure 8 reveals that the final-age strengths range from 70–90 MPa. Figure 8 reveals that *R* and *w* values increased as the final-age strength values increased. Figure 8b examined the effect of *R* and *V* values on final-age strength. Similar results were obtained here, and an increase in *R* and *V* values was observed as the final-age strength increased. Figure 8c shows the effects of *V* and *w* parameters on final-age strength.

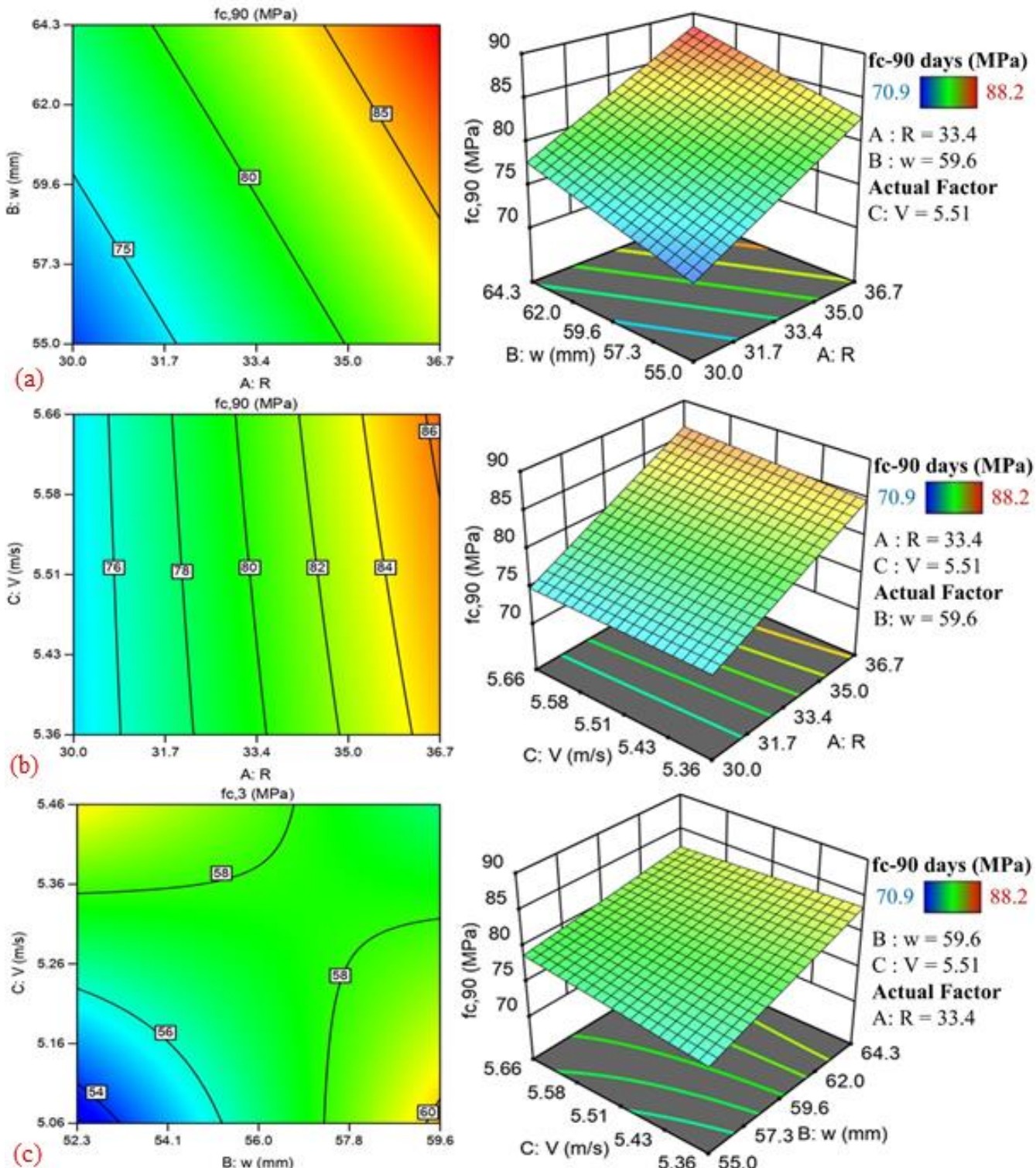

**Figure 8.** The effect of model 4 on final-age strength: (**a**) Model 1, (**b**) Model 2, and (**c**) Model 3.

### 3.2.3. Evaluation of Developed Models

In order to evaluate the effectiveness of the four developed mathematical models, the early and final-age strength results and the prediction results of the models were evaluated in detail. The prediction results, absolute relative deviation (ARD) values, and early- and final-age strengths of the developed models are shown in Figure 9. ARD values were calculated using Equation (1).

$$\text{ARD (\%)} = \frac{\text{Experimental} - \text{Model}}{\text{Experimental}} * 100 \qquad (1)$$

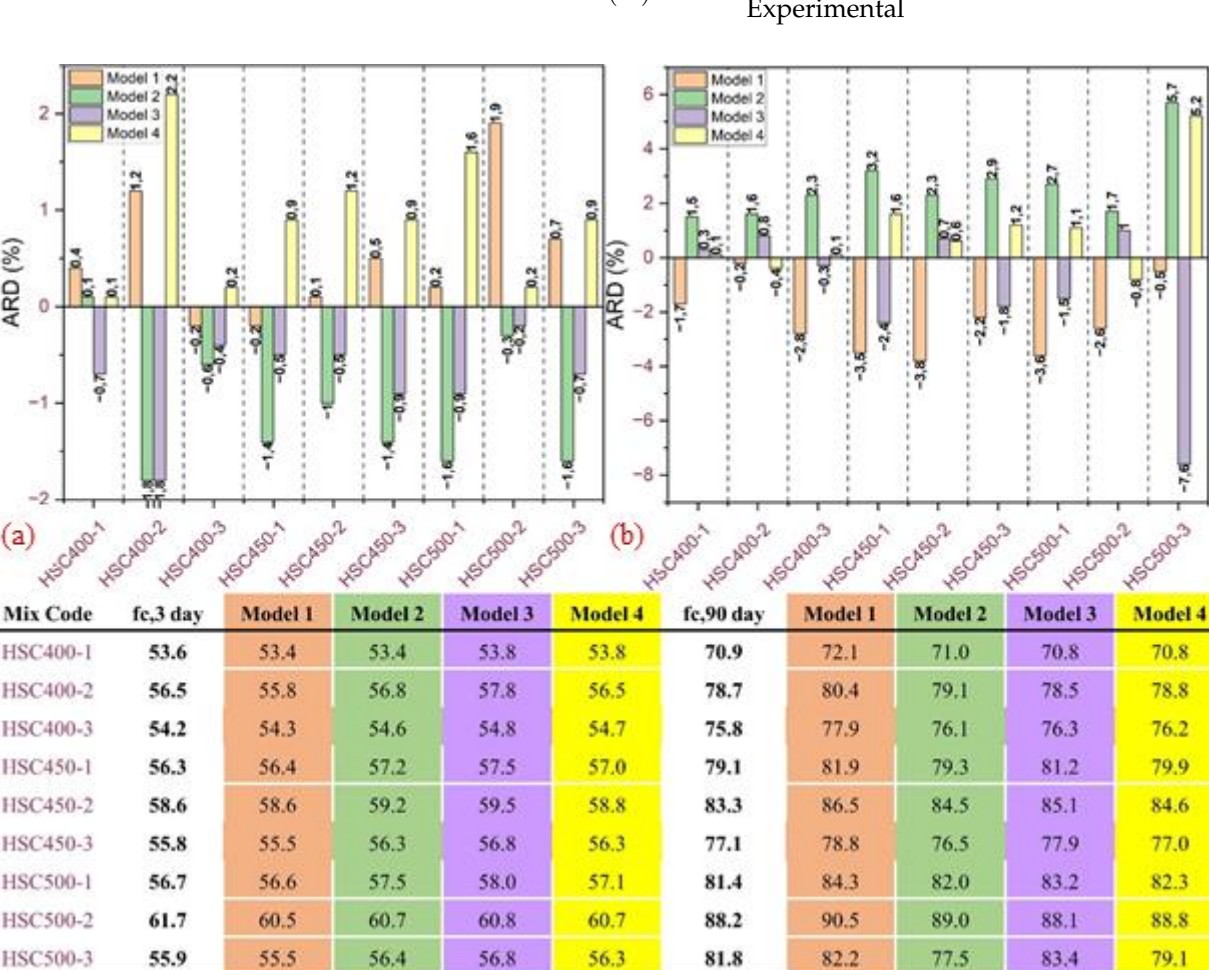

| Mix Code | fc,3 day | Model 1 | Model 2 | Model 3 | Model 4 | fc,90 day | Model 1 | Model 2 | Model 3 | Model 4 |
|----------|----------|---------|---------|---------|---------|-----------|---------|---------|---------|---------|
| HSC400-1 | 53.6 | 53.4 | 53.4 | 53.8 | 53.8 | 70.9 | 72.1 | 71.0 | 70.8 | 70.8 |
| HSC400-2 | 56.5 | 55.8 | 56.8 | 57.8 | 56.5 | 78.7 | 80.4 | 79.1 | 78.5 | 78.8 |
| HSC400-3 | 54.2 | 54.3 | 54.6 | 54.8 | 54.7 | 75.8 | 77.9 | 76.1 | 76.3 | 76.2 |
| HSC450-1 | 56.3 | 56.4 | 57.2 | 57.5 | 57.0 | 79.1 | 81.9 | 79.3 | 81.2 | 79.9 |
| HSC450-2 | 58.6 | 58.6 | 59.2 | 59.5 | 58.8 | 83.3 | 86.5 | 84.5 | 85.1 | 84.6 |
| HSC450-3 | 55.8 | 55.5 | 56.3 | 56.8 | 56.3 | 77.1 | 78.8 | 76.5 | 77.9 | 77.0 |
| HSC500-1 | 56.7 | 56.6 | 57.5 | 58.0 | 57.1 | 81.4 | 84.3 | 82.0 | 83.2 | 82.3 |
| HSC500-2 | 61.7 | 60.5 | 60.7 | 60.8 | 60.7 | 88.2 | 90.5 | 89.0 | 88.1 | 88.8 |
| HSC500-3 | 55.9 | 55.5 | 56.4 | 56.8 | 56.3 | 81.8 | 82.2 | 77.5 | 83.4 | 79.1 |

**Figure 9.** Experimental and predicted results: (**a**) Early-age, (**b**) Final-age strength.

Figure 9 shows the developed models' prediction results, ARD values, and early- and final-age strengths. A detailed assessment of Figure 9 shows that all ARD values of the developed models for both early- and final-age strength were below 10%. It was noteworthy that all of these values were below 5%, especially for early-age strength. Examining the ARD values belonging to the developed models illustrates that the mathematical models that predicted the early- and final-age strengths with the highest accuracy were model 1 and model 4, respectively. Model 1 made highly accurate predictions of early-age strength values, and ARD values for almost all concrete series remained below 1%. It was also seen that almost all of the ARD values of model 2 were below 1%. The ARD values for the final-age strength were higher, but it was noteworthy that all of the ARD values obtained were below 10%. A detailed examination of Figure 9b shows that model 4 predicted all concrete series with very high accuracy except the HSC500-3 series. Examination of model 1 reveals satisfactory results when estimating the final-age strength. It was also noteworthy that all ARD values of model 1 were below 5%. The common point of these two models

was that they consider the $R$ and $w$ values. In addition, a detailed evaluation of all models revealed that all the developed models were quite successful in predicting both early- and final-age strength. However, it was determined that models containing $V$ parameters were one step behind compared to other models. The main reason for this situation was the sensitivity of the $V$ parameter to the moisture content.

## 4. Conclusions

Determining the early- and final-age strength is of great importance in determining the concrete's quality, the formwork stripping time, and, indirectly, the project duration. For this purpose, combined methods were developed to predict both early- and final-age strength with high accuracy by using binary and ternary non-destructive test methods without damaging the structural element. The results and recommendations of the study are given below:

- Absolute relative deviation values of all the developed models were below 10% for both early- and final-age strength, and the effectiveness of the models was observed in detail.
- Early-age strength significantly affects the formwork stripping time and, indirectly, the project duration. Model 1 predicted the early-age strength with very high accuracy, and all absolute relative deviation values were below 2%. It is thought that the use of this model in constructions where early age concrete strength and formwork stripping time must be fast will provide significant benefits. However, since the Windsor probe test is a costly test method due to the single use of each probe, the selection of the usage area is also of great importance.
- Model 4 predicted the final-age strength with the highest accuracy. This model includes three different non-destructive test methods, in contrast to the combined methods developed by using two non-destructive testing methods together in the literature. It was determined that the combined methods developed by using three different non-destructive test methods for the final-age strength gave better results than the double combined methods.
- Among the developed models, it was seen that both the early- and final-age prediction accuracy of model 3 were lower than the other models. The main reason for this situation was the ultrasonic pulse velocity test considered in the model. Since this test method was highly sensitive to moisture content, the prediction results were lower than other models. Care should be taken to ensure that the surfaces of the concrete samples to be estimated using this model are dry and that the moisture content is low.
- While the repetition of the non-destructive test methods does not damage the carrier elements, the repetition of the destructive test methods causes great damage to the carrier elements. For this reason, using non-destructive test provides great advantages in many respects. Therefore, it is thought that the combined methods to be developed using different non-destructive test methods will greatly benefit the literature and the construction sector. Different combined methods developed for this purpose and their usage areas were examined in detail.
- Due to the many newly constructed buildings since the 24 January 2020 Sivrice-Elazig earthquake, rapid progress of the construction, the determination of the concrete quality with high accuracy, and the delivery of the project at the desired time are of great importance. Therefore, it is thought that determining the strength values with high accuracy using the developed combined methods without damaging the building element will provide benefits in terms of time, labor, and cost.

**Author Contributions:** T.D. and M.U. carried out experimental studies, production of samples, and implementation of tests in the laboratory. M.U. also contributed to the writing, design, and interpretation of all sections of the article. K.E.A. reviewed all parts of the article and made significant contributions to eliminating deficiencies. All authors have read and agreed to the published version of the manuscript.

**Funding:** This research is supported by the Scientific Research Project Fund of Firat University under the project number MF.21.51. There is no funder.

**Data Availability Statement:** Data and materials will be made available on request by authors.

**Conflicts of Interest:** The authors declare no conflict of interest.

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
