# Peer review of "Development of Combined Methods Using Non-Destructive Test Methods to Determine the In-Place Strength of High-Strength Concretes"

_processes, doi:10.3390/pr11030673_

Round 1

Reviewer 1 Report

This manuscript reports the developing of combined NDT methods to test high-strength concrete. The study is well designed and executed but I still have some basic questions about the combination of different NDT methods. I am suggesting major revision for the manuscript. Further revision is needed before publication.

My questions/suggestions are as follows-

1.      Author should clearly mention the motivation and novelty of the present work compared to the previous studies on the use of different combination of NDT methods. The authors should put more in context their findings and spell out clearly how their results could be important for further progress of the related fields and why rather than simply stating it every now and again.

2.      As the topic is too specific, authors should present the findings in more compact manner to access the attention of the broad community. Somehow the interpretation was limited to only support the claim on using different combinations of NDT methods. It is important to discuss data in totality, rather than focusing on only the parameters supporting the proposed hypothesis. More detailed and focused analysis must be performed to understand this curiosity that this paper highlights.

3.      How accurate the results of different NDT models in this paper compared to DT methods?

4.      Is dual or triple NDT methods time and cost effective?

5.      Which model the author would recommend as most effective w.r.t. benefits, time, and cost and why?

6.      How many times each sample was measured to extract different parameters? Put an error bar in each point.

7.      Is the sequence of measuring (e.g., ultrasonic pulse velocity first then others) has an effect on results?

Author Response

The file has been prepared by considering the values, suggestions and comments of the referee. It is presented in the appendix.
best regards

Reviewer 2 Report

The paper combines 3-4 NDT methods to determine the strength of concrete, the paper does not present any novel methodology or any scientific findings which can be useful to the readers

O. Tsioulou et al., 2017, already presented Combined non-destructive testing (NDT) method for the evaluation of the mechanical characteristics of ultra high performance fibre reinforced concrete, here the authors have to mention how their HSC performance is different and what different effect it can be when using NDT.

If more than three NDT methods were combined it would have been more accurate, that doesn’t mean you can go on combining all the methods available.

The authors should present the scientific reason for combining two or three methods which one one method is not able to provide, atleast they should have done some real life studies which proves that using one method doesn’t provide accurate results and the reasons for the same.

Line 278-290, the strength value increased with the r & V value it can been seen in Fig.4 also before combining????

If the combined model error was below 10% what is the error individually???

Line 373, Model 4 predicted the final strength accurately not the early strength , authors should give the reasons for the same

Line 379, Model 3 values are less than others because of UPV, so does the authors conclude that UPV is not suitable???to combine with any other methods??

But individually UPV results corroborate with the strength Fig.4, then when combined why the model is not predicting accurately??

Author Response

(The authors gave the same response as above.)

Reviewer 3 Report

Overall, the study is interesting and worthy of investigation. The following comments should be addressed.

 1.      Provide Graphical Abstract.

2.      Abstract: the abstract need to be summarized the main points and avoid the unnecessary parts to better understanding and readability.

3.      Abstract: Please underscore the scientific value added in the abstract. Add some of the most critical quantitative results to the Abstract.

4.      In the introduction section, the aim of the paper should be presented clearly. Additionally, certain results obtained from the presented study should be given in the last paragraph of the introduction section.

5.      What is the hypothesis behind this work should be articulated?

6.      Samples sizes are inadequate.

7.      Discussion of the result section is not understandable. Needs improvements in the discussion section of each property.

8.      The presentation of the study conclusions should be improved.

  1. All changes must be highlighted in the revised manuscript.

Author Response

(The authors gave the same response as above.)

Round 2

Reviewer 1 Report

The authors gave reasonable answers of my concerns. I am happy to recommend the manuscript to be published as is.

Reviewer 2 Report

The authors have answered the questions raised